# Content, Mechanism, and Outcome of Effective Telehealth Solutions for Management of Chronic Obstructive Pulmonary Diseases: A Narrative Review

**DOI:** 10.3390/healthcare11243164

**Published:** 2023-12-14

**Authors:** Saeed Mardy Alghamdi

**Affiliations:** Respiratory Care Program, Clinical Technology Department, Faculty of Applied Medical Sciences, Umm Al-Qura University, Makkah 21961, Saudi Arabia; smghamdi@uqu.edu.sa

**Keywords:** digital health, telehealth, COPD, review

## Abstract

Telehealth (TH) solutions for Chronic Obstructive Pulmonary Disease (COPD) are promising behavioral therapeutic interventions and can help individuals living with COPD to improve their health status. The linking content, mechanism, and outcome of TH interventions reported in the literature related to COPD care are unknown. This paper aims to summarize the existing literature about structured TH solutions in COPD care. We conducted an electronic search of the literature related to TH solutions for COPD management up to October 2023. Thirty papers presented TH solutions as an innovative treatment to manage COPD. TH and digital health solutions are used interchangeably in the literature, but both have the potential to improve care, accessibility, and quality of life. To date, current TH solutions in COPD care have a variety of content, mechanisms, and outcomes. TH solutions can enhance education as well as provide remote monitoring. The content of TH solutions can be summarized as symptom management, prompt physical activity, and psychological support. The mechanism of TH solutions is manipulated by factors such as content, mode of delivery, strategy, and intensity. The most common outcome measures with TH solutions were adherence to treatment, health status, and quality of life. Implementing effective TH with a COPD care bundle must consider important determinants such as patient’s needs, familiarity with the technology, healthcare professional support, and data privacy. The development of effective TH solutions for COPD management also must consider patient engagement as a positive approach to optimizing implementation and effectiveness.

## 1. Introduction

Chronic obstructive pulmonary disease (COPD) is a common, progressive respiratory disease that is growing in prevalence worldwide [1]. The Global Initiative for Chronic Obstructive Lung Disease (GOLD 2023) defines COPD “*as a heterogeneous lung condition characterized by chronic respiratory symptoms (dyspnea, cough, expectoration and/or exacerbations) due to abnormalities of the airways (bronchitis, bronchiolitis) and/or alveoli (emphysema) that cause persistent, often progressive, airflow obstruction*.” [2]. GOLD is a well-known international organization, which is working to provide guidelines for COPD diagnosis, treatment, prevention, and management in order to assess the characteristics of COPD at a global scale, and to maximize available resources to support COPD care [1]. In COPD management, pharmacological and non-pharmacological treatments are mostly administrated together [3]. Pharmacological treatments such as bronchodilators, expectorants, antibiotics, and inhaled corticosteroids assist in controlling symptoms [3]. In parallel with non-pharmacological treatments, such as smoking cessation, pulmonary rehabilitation (PR), and vaccination, which play a role in self-management as well as maintaining and controlling the condition [4]. Technology can play an important role in facilitating the delivery of these treatments, promoting self-management. Even with the acute exacerbation of COPD (AECOPD), the “acute worsening of respiratory symptoms that results in additional therapy”, patients may benefit from using technology to monitor the disease or detect the onset of deterioration.

Telehealth (TH) solutions for COPD care are promising therapeutic interventions and are valuable in helping individuals to manage symptoms, minimize hospitalization, and improve their health status [5,6]. Also, the current evidence supports self-management training via TH as a crucial component of the COPD care bundle to help individuals change their behaviors, thus controlling their symptoms and reducing hospital readmission due to AECOPD [7,8]. Since TH solutions for COPD care first appeared, researchers across the world have been able to better understand how to create, implement, and maintain TH solutions. Although, as of October 2023, more than 184 clinical trials documented the use of TH solutions with COPD, there are no data establishing an overview of the experience with TH solutions. In addition, there is a lack of knowledge in determining the connection between the content, mechanism, and outcome of these using interventions when dealing with COPD patients. To address this gap in the knowledge, a narrative review was conducted with the aim of summarizing the existing literature about the structured TH in COPD care as well as informing future developers about the current status.

## 2. Methods

The keywords related to telehealth and digital health solutions for COPD management were searched in two databases (PubMed and Medline) up to October 2023. The content, mechanism, and outcomes of clinical trials introducing and providing TH or digital health solutions were included. The review excluded articles that were not published in English or were not related to the scope of the review. To ensure that our review included appropriate and necessary keywords, we added Medical Subject Headings to the search as well as working with the health sciences librarian. Potential articles from the databases were exported to EndNote 20. After removing duplications, the main author reviewed all titles and abstracts and applied eligibility criteria. To collect data, a standard Excel spreadsheet was created. The final findings are summarized and criticized in narrative form. An example of the search strategy is provided in Appendix A.

## 3. Results

Research findings included 30 clinical trials that reported or provided TH or digital health solutions to manage COPD patients (Table 1). The oldest clinical trial was reported in 2008 and the latest was reported in 2023. Data were summarized and criticized, with a special focus on articles reporting details regarding interventions, including the content, mechanism, and outcomes.

### 3.1. Digital Solutions in Healthcare Services

The term TH became the most common terminology to describe this integration of information and communication technologies (ICTs) and healthcare services. It is currently the preferred term because it captures the broad applications of this technology in providing healthcare services [9]. The term digital health is a relatively new concept, defined as “the use of digital technologies for health” [10]. Both TH and digital health are broad concepts that might include planning, monitoring, assessment, diagnosis, education, and treatment [10]. In the literature, TH or digital health have been used successfully within different health care disciplines, including cardio-respiratory disease management, and in the provision of home care or self-management for patients with chronic diseases such as COPD [11,12]. In addition, technology can be used in COPD care to improve the timeframe of therapeutic contact, or as an alternative tool when access to care is not available [13] Figure 1.

### 3.2. Benefits of TH with COPD Management

Systematic reviews and meta-analysis showed that TH solutions that support self-management could contribute to improve user’s skills in controlling COPD, especially those with transport or economic limitations, or geographical barriers [12,14]. Considering the high cost of health care services, the TH approach could lower costs while maintaining quality of treatment in primary care clinics, decrease the pressure on the specialty care clinics, deliver tailored care, and facilitate the coordination of care among healthcare professionals [15]. Also, TH could be an opportunity to facilitate education reinforcement for professionals, patients and/or caregivers [16]. Additionally, digital health allows for frequent reminders to be provided to users to practice behavior which strengthens an existing behavior, like self-monitoring and self-efficacy, or facilitates early interventions for COPD conditions by detecting their exacerbation at an early stage, and could therefore prevent unnecessary emergency visits and hospital readmissions [17,18], as shown in Figure 2.

### 3.3. Structures of TH Solutions

Development and innovation in digital health are steadily increasing [19]. Looking to digitalization alone as effective tool is not going to provide us with a better result in clinical practice. As clinicians, we could look at this technology as a key part of the package provided to the users. The package usually contains multiple components, which include complex mediums and interventions (i.e., telerehabilitation). However, it is difficult to evaluate the functionality of TH solutions that support disease management without considering three important components: (1) the context that was provided, (2) the mechanisms that package components went through, and (3) the outcomes that the solutions are targeting. The answers to these questions still need further clarification [20,21]. One way to resolve this conflict is understanding the functionality of digital health solutions in COPD management [22,23]. This functionality could be understood from a theoretical perspective by identifying the associations between the context, mechanisms, and outcomes of current TH solutions that support disease management, in other words, the conceptual frameworks that enhance the connection between these elements and make technology a valuable tool in the delivery of care (Figure 3).

### 3.4. Context of Digital Health Solutions

The context of TH in COPD management includes three important factors: (1) the clinical setting, (2) digital health development, and (3) targeted populations. A summary of the literature is provided in Table 1.

**Table 1 healthcare-11-03164-t001:** TH interventions’ characteristics from 2008 to 2023.

Authors	Settings	Intervention with TH	Mode of Delivery	Intensity	Control Group Treatment	Summary of TH Outcomes Compared to Control Group
Trappenburg et al., 2008 [24]	Patient home (n = 165)	Software COPD self-management education	Telephone	1×/day for 6 months	Usual care	Reduction in AECOPD and hospitalization
Koff et al., 2009 [25]	Patient home (n = 40)	Online COPD self- management education	Web-based and telephone	1×/day for 3 months	Usual care	Improved quality of life
Halpin et al., 2011 [26]	Patient home (n = 79)	Automated text messages system	Text message	1×/day for 4 months	NR	Lower AECOPD but no change in quality of life
Lewis et al., 2011 [27]	Patient home (n = 40)	Home monitoring	Telephone	2×/day for 6 months	Usual care	No difference in hospitalization or length of stay
Stickland et al., 2011 [25]	Community center (n = 409)	Online COPD self- management education	Web-based and video calls	2×/week	In-person rehabilitation	Improvements in quality of life
Antoniades et al., 2012 [9]	Patient home (n = 44)	Online COPD self-management education	Web-based	1×/day for 12 months	Usual care	No reduction in hospitalization or improvement in quality of life
Chau et al., 2012 [28]	Patient home (n = 40)	Telecare services	Web-based and telephone	3×/day for 2 months	Community services	No difference in health-related quality of life
Dinesen et al., 2012 [29]	Patient home (n = 111)	Telerehabilitation	Video calls	1×/day for 4 months	Instructional book	Reduced hospitalization
Nield et al., 2012 [30]	Patient home (n = 22)	Online COPD self- management education	Video calls	1×/week for 1 month	Usual care and in-person education	Decreased dyspnea
De San Miguel et al., 2013 [31]	Patient home (n = 80)	Written COPD self- management education	Telephone	1×/day for 6 months	Usual care	Reduced hospitalization and length of stay
Pedone et al., 2013 [32]	Patient home (n = 99)	Telemonitoring	Web-based, telephone, and algorithm	1×/day for 9 months	Usual care	Reduced respiratory events and hospitalization
Pinnock et al., 2013 [33]	Patient home (n = 256)	Online COPD self- management education	Web-based, telephone, and algorithm	1×/day for 12 months	Usual care	Reduced admission to hospital but no change in quality of life
Schou et al., 2013 [34]	Patient home (n = 44)	NR	Video calls	1× /day for 3 months	Regular Hospitalization.	Improvements in lung volumes and oxygen saturation
Calvo et al., 2014 [17]	Patient home Medical center (n = 60)	Home care with TH	Web-based and telephone	1×/day for 7 months	Usual care	Reduction in ER visits, hospitalization, and length of stay
Tabak et al., 2014 [35]	Patient home (n = 29)	Online COPD self- management education	Web-based and Video calls	1×/day for 9 months	Usual care	Increased patient adherence to exercise
Berkhof et al., 2015 [36]	Medical center & patient home (n = 101)	Phonecalls, education and follow ups	Telephone	Call/2 weeks for 6 months	Usual care	No improvements in health status
Jakobsen et al., 2015 [6]	Patient home (n = 57)	Virtual hospital	NR	1×/day for 6 months	Regular hospitalization	Reduced re-admission to hospital due to AECOPD
McDowell, 2015 [37]	Patient home (n = 110)	Home-based health care	Telephone	1×/day for 6 months	Usual care	Improved health related quality of life.
Ringbeak et al., 2015 [38]	Patient home (n = 281)	Online COPD self- management education and home exercise	Video calls	3×/week for 6 months	Usual care	No change in dropout or mortality
Tucker et al., 2016 [39]	Patient Home (n = 65)	Written home exercise with TH	Telephone	Call/2 weeks	Usual care	Improvement in physical activity
Ho et al., 2016 [40]	Patient home (n = 106)	Telemonitoring	Web-based and telephone	1×/day for 2 months	Usual care	Reduced number of hospitalizations due to AECOPD
Ringbeak et al., 2016 [41]	Patient home and outpatient (n = 116)	Online COPD self- management education and home exercise	Web-based, video calls	NR	In-person rehabilitation	Improved physical capacity but no improvement in CAT score.
Vianello et al., 2016 [42]	Hospital (n = 334)	Online COPD self-management education	Web-based and telephone	1×/day for 12 months	Usual care	Reduced readmission rate due to AECOPD
Farmer et al., 2017 [43]	Patient home (n = 116)	COPD self-management education with TH	Web-based	1×/day for 12 months	COPD self- Management education without TH	Improved health status and quality of life
Lilholt et al., 2017 [44]	Patient home and community center (n = 1225)	Telerehabilitation	Web-based, and telephone	NR	In-person rehabilitation	No difference in quality of life
Shany et al. ,2017 [45]	Patient home (n = 42)	Online COPD self- management education	Web-based and telephone	1×/day for 12 months	Home care	Reduction in hospitalization and length of stay
Tsai et al., 2017 [46]	Patient home (n = 37)	Online COPD self- management education and home exercise	Video calls	3×/week	Usual PR	Improvements in physical capacity and quality of life
Soruano et al., 2018 [47]	Patient home (n = 237)	Telemonitoring	Internet modem	1×/day for 12 months	Usual care	TH did not reduce hospitalization due to AECOPD
Jolly et al., 2019 [48]	Patient home (n = 58)	Multimedia COPD self- management education and telephone coaching	Telephone	1×/day for 12 months	Usual care	Improvement in uptake in PR program
Jiang et al., 2020 [49]	Patient home (n = 106)	TelePR program	WeChat	3×/week for 6 months	Usual PR	No difference in symptoms score between TelePR and UC
Rassouli et al., 2021 [50]	Patients home (n = 168)	Online COPD self- management education	Web-based and telephone	5×/week for 6 months	Usual care	TH use improved CAT score and satisfaction with care.
Zanaboni et al., 2022 [51]	Patients home (n = 120)	TelePR	Video calls	3×/week for 2 years	Treadmill at home	TH redued hospitalization.
Polo et al., 2023 [52]	Hospital and participants’ home (n = 209)	COPD TelePR program	Zoom and web-conferencing	2×/week for 2 months	Usual PR	TH improved COPD symptoms, fatigue, self- management, and lung volumes.

TH: telehealth, COPD: chronic obstructive pulmonary disease, PR: pulmonary rehabilitation, NR: not reported, TelePR: telepulmonary rehabilitation, AECOPD: acute exacerbation of COPD, ER: emergency room, CAT: COPD assessment tool, x: times, n: total number of participants in the study.

#### 3.4.1. Clinical Settings

TH solutions have potential benefits in primary and secondary clinical settings [53]. For example, in the primary clinical settings, TH facilitates communication between the healthcare professionals and the patients to provide effective monitoring and assessment between clinical visits [54]. This application has been found to be helpful in clinical settings to provide prompt feedback and enhance exercise and treatment adherence [54]. In the secondary clinical settings, TH is used to facilitate coordination of care between primary and secondary health care centers, especially in small modern cities [55,56]. The coordination of care allows for healthcare providers like specialists to assess the COPD patients and provide advice on the action plan without traveling to the patient’s location [56,57]. Mostly, this kind of TH solution refers to teleconsultations [58]. Clinical settings, for COPD, must consider setup and logistics as we are witnessing an increased interest in shifting routine care to an entirely digitalized setup. This phenomenon was accelerated by circumstances such as disease outbreaks [59,60] and some strategic plans to decrease the cost burden of these diseases [61], as shown in Table 2.

#### 3.4.2. Digital Health Solutions Development

According to the *Consolidated Standards of Reporting Trials of Electronic and Mobile Health Applications and onLine TeleHealth* (CONSORT-EHEALTH), TH interventions must be reported as one of three categories [62]. First, as an educational tool to provide education and improve patient knowledge, such as “living well with COPD or Asthma+Lung UK” [63,64]. Second, as an electronic tool for communication and to facilitate home-monitoring and/or remote monitoring for individuals with COPD. Third, as a comprehensive TH intervention to provide education, monitoring, assessment, and treatment to help manage the condition, such as telepulmonary rehabilitation programs and virtual hospitals [6,34]. The development of TH solutions in COPD also must consider patient engagement as positive approach toward optimizing implementation and improving usability [65]. The current TH interventions have introduced a variety of potential benefits that guided their development. None of the studies included in this review reported the theoretical framework that guided the development of the TH intervention.

### 3.5. Mechanism

TH solutions with COPD are multicomponent interventions (Table 3). Not all the TH solutions involved interventions [66]. There are studies of TH tools in the literature that only provided remote monitoring for COPD without any further interventions, even self-management or education [36,67], while others provided a virtual hospital at the patient’s home [6,34]. The mechanism of TH solutions is complicated, and depends on professional support as well as the standard care practice. To describe the mechanism of TH solutions that supported COPD management, we must consider four important factors: (1) content and format, (2) modes of delivery and level of technology, (3) strategy, and (4) the intensity of remote monitoring, which includes its frequency and duration, as shown in Figure 4.

#### 3.5.1. Content

The content of TH solutions is a broad topic. The evidence presented a wide range of variation in the content of COPD management [12,21]. That variation in the content comes from the different perspectives regarding the self-management support and/or different guidelines used to manage COPD. For instance, self-management with COPD has been defined as individuals’ ability to master a broad range of tasks, including managing their symptoms, physical activity, coping with the mental consequences, and maintaining their lifestyle [87,88]. This support flows through a continuous opening channel via TH solutions and could include periodic home visits or calls to follow-up with the patients and provide feedback [89].

First, in symptoms’ management using TH solutions, the content was described in the literature as giving the patients instructions, advice, and recommendations regarding what actions must be taken to control symptoms [36,45,79]. Patient–professional communication or professional–professional communication were used to visualize and monitor symptoms [36,45,79]. This supported disease maintenance and could help both patients and professionals to detect deterioration at an early stage [9,36,90].

The second type of content is physical activity via TH solutions. Physical activity has been described as giving the patient instruction, advice, and recommendations on how to perform exercise and training [38,46,80,81]. The exercise includes one or a combination of the following components: (1) structure physical exercise, (2) techniques for performing daily activity, (3) set exercise, (4) stretching and legs exercise, and (5) techniques for conserving energy. This content is mostly used with stable COPD patients without any recent exacerbations [38,46,80,81].

The third content is mental and well-being support. The mental and well-being support has been described as giving the users instructions, advice, and recommendations regarding psychological support [24,25,30,33,82,83]. The psychological support could include one or a combination of behavior change techniques [23]. These are mostly delivered to the patient in-person by the healthcare providers, especially when TH systems recognize alarms from the patient’s side. The healthcare providers contact the patients and reassure them, provide information or feedback, and recommend relaxation techniques [23].

When these contents are delivered via TH solutions, a robust mechanism associated with these contents is required to explain how the solutions support feasibility and accessibility to improve health outcomes among COPDs [21,55]. Despite the variety in the content of TH solutions with COPD, all of them used either text or videos format. The text format could be delivered in many different modes, such as instruction books, written guidelines, and SMS messages [29,36,85], while the video format could be used to coach or teach the users a specific exercise, and/or provide recorded education and instructions [30,43,46].

#### 3.5.2. Mode of Delivery

The modes of TH solutions in COPD care have been defined as the way that the content is delivered to the end-users [91]. Since the technology is a fast-growing field, there are three different styles of delivery, but in this paper we considered ICT as a crucial mode of delivery. From this perspective, TH support has been summarized as synchronous, asynchronous, and immediate or live analytics and decisions [84,92]. Advanced forms of synchronized digital health support via machine learning (ML) and artificial intelligence (AI) are growing and getting more attention. These involve using existing data to develop and train statistical models to support clinical judgment. The simple version of this method is described as reviewing the patterns in the data to reach an informed conclusion. It can be used to diagnose or treat people with COPD. Establishing these methods in COPD care requires a lot of data, and these methods are still under review by experts [92].

#### 3.5.3. Strategy of Delivering Digital Health Support

Individual strategy

The best example to describe this strategy is mobile health apps [9,43,63]. In terms of interaction, the users must access the mobile app to receive feedback from the healthcare provisional regarding their conditions, or feedback from a predesigned algorithm [93,94]. In terms of technology, this strategy requires advanced technological devices with internet access (i.e., a smart cellphone or tablet) to visualize, send, or receive information. This strategy involves partial interaction between the users and the healthcare providers, but full interaction between the users and the technological devices. For example, feedback could be delivered to the patient using well-known platforms such as “Living well with COPD” [95] and “myCOPD” [96] These platforms allow for the patient to access the education program and could be used at any time [63,95,96,97]. Another proposed approach to the individual strategy includes using a platform designed based on local guidelines. This would require the patient to access the platform to conduct self-monitoring and receive education, support, and feedback from their healthcare provider [43]. Both approaches enhance the personalized care for individuals with COPD [98].

Group strategy

The best example of this strategy is providing services in regular meetings, also known as community-based TH support [29,36,85,86]. In terms of interaction, this strategy involves interaction between users and healthcare professionals. This strategy does not require advanced ICT, but requires more effort from the patient and healthcare provider to complete the session or meet competencies [99]. The group strategy requires qualified healthcare professionals to deliver services or care. The patient, on the other hand, must attend the clinical or community center to receive support such as community-based telePR for COPD [99].

### 3.6. Intensity

#### Duration and Frequency

Treatment or management sessions for COPD via TH solutions have different intervals and durations. For example, some studies provide COPD management sessions for 10–20 min per day, mostly involving pulmonary rehabilitation and breathing exercises [26,27,46,100]. It seems that a specific aspect of self-management via TH (i.e., breathing exercises and pursed-lip breathing) takes less time than comprehensive disease management (i.e., exercise-coaching, energy conservation, medication follow-up and nutritional advice, and managing stress). Regarding the frequency of sessions, daily or weekly frequencies for a short duration seen more frequent in the literature [12,101].

### 3.7. Health Outcomes

TH solutions that support COPD management have variety of outcomes. This variety might come from the inconsistencies in using a theoretical model when the researchers design a TH solution [102]. For example, if a researcher designs a study to understand the rate of acceptance of digital health solutions among COPD patients, then the outcomes measured for that study must be related to the rate of technology acceptance, such as perceived ease of use and perceived usefulness [103]. TH studies in the literature targeted various outcomes, including clinical outcomes, treatment cost or adherence, and changes in patient behavior [104,105,106]. Also, clinical outcomes such as time to AECOPD, emergency department visits, hospitalization [107], need for non-invasive mechanical ventilation [108], length of stay [109], physical activity [110], health status, and HRQoL [111] were mostly measured as primary or secondary outcomes [106].

### 3.8. Determinants of TH Solutions

Before implementing TH solutions, stakeholders and policymakers must think about the factors that influence adaption and adherence. Providing individuals with easy access to basic and advanced communication technologies, as well as digital health platforms, is an important determinant. Another factor is the presence of well-trained healthcare experts to manage and administer the platform, as well as a technical support team available 24 h a day, 7 days a week to resolve any technological issues. Third, data transformations on the envisaged digital health platforms must be safe. Fourth, platform users’ needs must be taken into account, such as hearing and vision limitations in the elderly [14].

### 3.9. Other Promising Applications for TH in COPD Care

Digital health can be used to provide medical education and training (tele-education), especially when there has been a disruption in direct management, training sessions, seminars, and medical education. Even in units like ICUs, tele-education and/or remote training via ZOOM, Skype, or Microsoft Teams has supplanted traditional techniques in teaching and training programmes [112]. In the COPD care literature, this experience has a positive influence and leads to better treatment adherence outcomes [52].

The use of virtual ICUs in management has rapidly increased to address the dearth of health care workers and to ease family visits to critically sick patients. The use of virtual ICU had positive results in terms of reducing physiological stress and anxiety, and enhancing staff morale, according to cross-sectional data [60]. In addition, virtual ICU is an excellent opportunity for family members to share their joy and relief [60]. Virtual reality (VR) has also played a significant role in the treatment of COPD patients. One of the most important applications of VR is in pain management for COPD patients with back pain. Experts have proved that using virtual reality to provide care to those people is a workable and powerful tool in healthcare delivery. In addition, VR can be a fun tool to stimulate physical activity among persons in PR programmes [14]. VR is extremely beneficial since it allows for these patients to continue their treatment at home, avoiding the need to visit community centres and hospitals.

AI is also considered an innovative new form of technology that has helped in COPD diagnosis [77]. Kaplen et al. evaluated the ability of AI to detect COPD diagnosis, and the results showed that AI can distinguish between COPD and asthma [77]. AI, therefore, can provide rapid information to make a medical decision.

There are certain limitations to this review. Firstly, we may not have included all prospective TH trials in this narrative review, because we only looked through two key databases: PubMed and Medline. Nonetheless, given that PubMed and Medline are considered the largest medical databases, we are certain that we have not overlooked any significant research in the literature. Although our research was restricted to publications released during the last 15 years, this evaluation encompasses a wide range of the literature.

## 4. Discussion and Conclusions

This review contributed to the literature by providing an overview of the existing structured TH solutions in COPD care and the way they work. Over the last 15 years, several studies dealing with this topic have been published. The current review concentrates on describing the connection between content, mechanism, and outcomes in clinical trials that provided TH solutions to individuals with COPD. This narrative review is the first to focus exclusively on this specific aspect of innovative care for COPD patients. Researchers have made significant effort to prove that TH approaches can increase the availability and accessibility of effective care for the COPD population. Despite this, many lessons have been learned from this review.

First, TH or digital health solutions in COPD management can be heterogeneous in their content, mechanism, and outcome, all of which have contributed to their efficiency and effectiveness. Second, current evidence suggests that using digital health solutions in COPD management could provide benefits such as managing symptoms and enhancing physical activity, and improving mental health. Third, some factors, including strategy, intensity, delivery medium, and content, can affect TH solutions’ function. Additionally, the three most commonly used outcome indicators for TH solutions were quality of life, health status, and treatment adherence. Therefore, the COPD care bundle’s successful implementation of TH must consider key factors such as patient demands, familiarity with the technology, support from medical professionals, and data privacy. Finally, to achieve optimal implementation and efficacy, theoretical frameworks and patient engagement must be considered in the development of TH solutions for COPD care.

Researchers should focus on developing smarter, interactive digital health solutions for people with COPD in the future. The integration of TH solutions into the model of care should consider simplification, as the technology could be a challenge for certain patients. It is recommended that future work report the degree to which the use of TH solutions has contributed to reducing healthcare costs. The principle of delivering high-quality healthcare services through various modes must be reinforced. This makes it possible for healthcare systems to use innovative approaches depending on the needs and preferences of patients.

## Figures and Tables

**Figure 1 healthcare-11-03164-f001:**
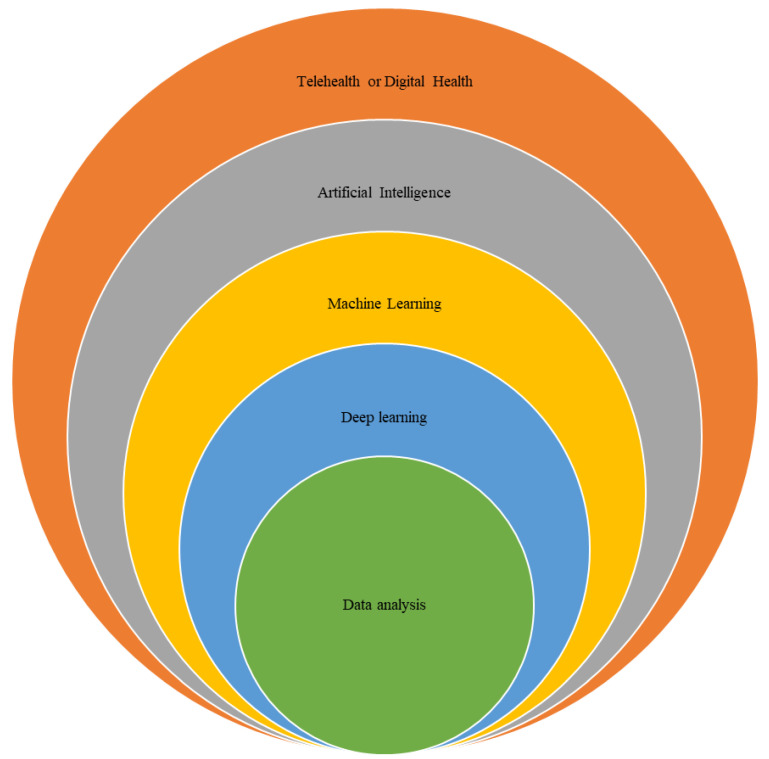
TH or Digital health concepts in COPD care.

**Figure 2 healthcare-11-03164-f002:**
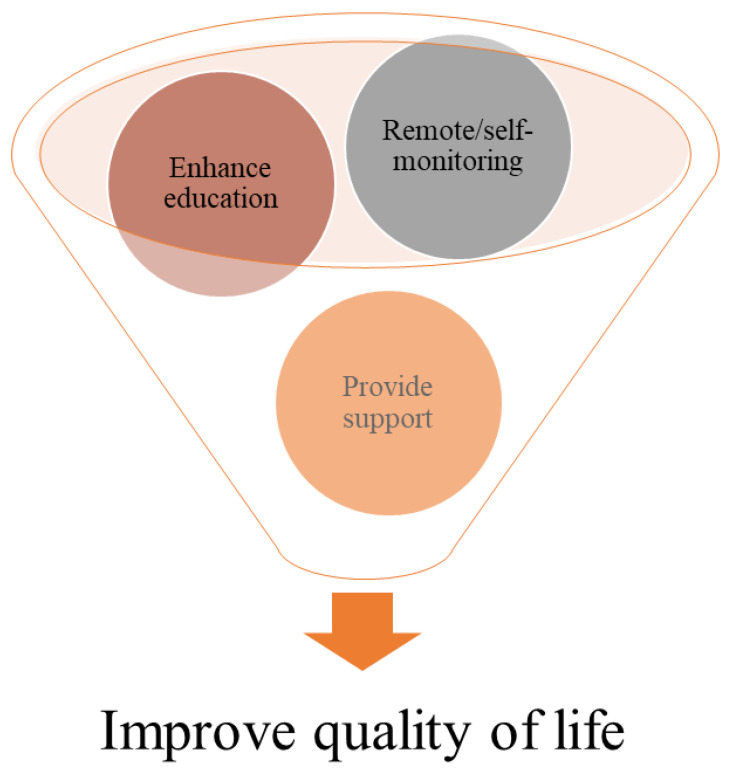
Benefits of TH with COPD management.

**Figure 3 healthcare-11-03164-f003:**
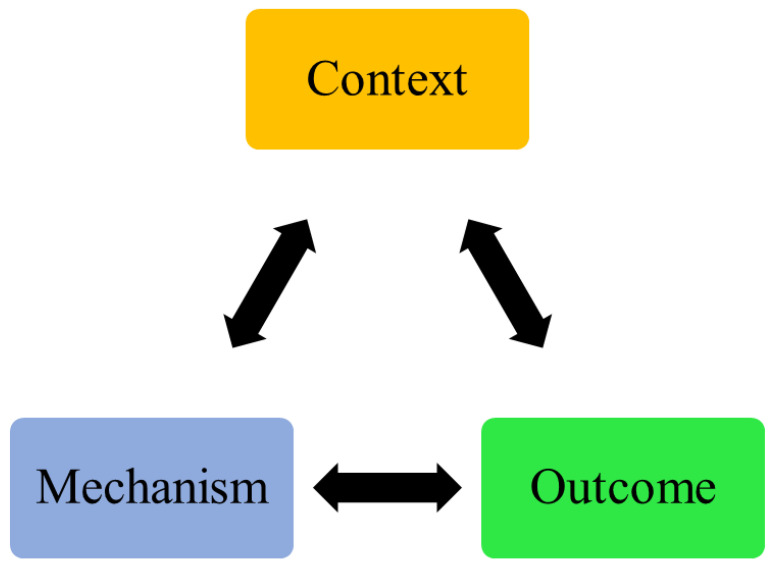
Structure of TH applications.

**Figure 4 healthcare-11-03164-f004:**
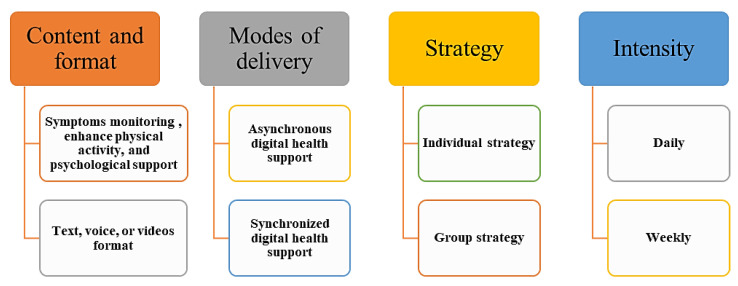
Factors describing the mechanism of TH solutions.

**Table 2 healthcare-11-03164-t002:** Applications of TH solutions in clinical settings.

Clinical Settings	Objective and Summary of the Potential Benefits
Primary care settings	Evaluate the acceptability and feasibility of real-time interactive voice and video calls via Skype for teaching breathing exercises and decreasing dyspnea compared to usual care. People who received the teaching via Skype reported less dyspnea than those receiving usual care [30]. Evaluate the effects of exercise coaching via digital health versus self-training without coaching in COPD. Ref. [46] Professionals provided the coaching in the study as part of the telerehabilitation program. The study found that exercise coaching via digital health helped COPD patients to increase their adherence and exercise capacity compared to self-coaching without digital health [46].
Secondary care settings	Determine the effects of The Health Buddy (HB) telemonitoring device on health consumption and health-related quality of life (HRQoL) in patients with moderate to severe COPD. Using HB resulted in decreased hospital days and outpatient visits, but no significant changes were observed in HRQoL [24]. Determine the effect of supervised, home-based, real-time videoconferencing telerehabilitation on exercise capacity, self-efficacy, HRQoL and physical activity in patients with COPD compared with usual care without exercise training. The study showed that telerehabilitation improved endurance, exercise capacity, and self-efficacy in COPD when compared with usual care [46].

Primary care setting defined as one service provided in primary care settings. Secondary care settings defined as more than one service provided by tertiary hospitals or community centers.

**Table 3 healthcare-11-03164-t003:** Summary of the mechanism factors of TH solutions.

Type of Support/Technology	Content (Covered One or a Combination of the Following Subjects)
▪ Special modems [68] ▪ Secure web-pages [69] ▪ Automated call/text center [70] ▪ Professional call center [71] ▪ Alert system to detect deterioration [72] ▪ Clinical decision software’s [73] ▪ Videoconference [46] ▪ Patients portals [74] ▪ Public internet platforms [75] ▪ Smartphone applications [76] ▪ Machine learning and artificial intelligence [77] ▪ ChatGPT [78]	▪ Symptoms management (i.e., breathlessness, medications, action plan) [36,45,79]. ▪ Enhance physical activity (i.e., exercise training, maintaining a healthy lifestyle, energy-conserving technique) [38,46,80,81]. ▪ Psychological and behavioral support (i.e., smoking cessation, adherence, compliance, problem-solving, self-efficacy, stress, and anxiety management) [24,25,30,33,75,82,83].
Mode of delivery	Strategy and intensity
▪ Synchronized and asynchronized TH support [84]: ○ Patient’s data stored in a secure modem. Then, COPD care provided by the healthcare professionals through digital health. ○ Digital systems recognize changes in data, then alert the healthcare center or healthcare team, but the system does not provide 24/7 communication (i.e., online chat or email) [84]. ○ Both modes considered as a means to store and forward digital health support [84]. ▪ Immediate and live analytics, support and decision-making are available 24/7 for the users (i.e., predesigned algorithms and ML), and this mode is considered a real-time digital health support [84].	▪ Strategy: ○ Individual or group [29,36,85,86]. ▪ Intensity: ○ Duration and frequency of treatment/data transmission [17,28,30,32].

## Data Availability

Not applicable.

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
