# Peer review of "Content, Mechanism, and Outcome of Effective Telehealth Solutions for Management of Chronic Obstructive Pulmonary Diseases: A Narrative Review"

_healthcare, 2023, doi:10.3390/healthcare11243164_

Round 1
Reviewer 1 Report
Comments and Suggestions for Authors
The manuscript addresses important topics about telehealth solutions for management of chronic obstructive pulmonary disease. The Abstract must be reviewed for the English language, consider rephrasing it. Line 118 change patient area for patient location. Line 148 Change ¨it does not necessary....... for It is not necessary. Line 149 should read......there are studies of TH tools in the literature that just provide. And so on.
Comments on the Quality of English LanguageConsider review for the English language.
Author Response
- Thank you for your valuable comments.
- The abstract has been rewritten. The revised manuscript has been reviewed by English language professional.
Reviewer 2 Report
Comments and Suggestions for Authors
1- The abstract needs to be edited because it lacks explanations
about the study method and findings and conclusions.
2- Telehealth is not synonymous with digital health, while in this study these two concepts were used interchangeably. Digital health is a type of electronic health based on artificial intelligence, mobile health, and molecular and cellular studies as well as precision medicine. It is suggested to use the term digital health because the focus of the study is on telehealth.
3- It is necessary to explain about the study method after the
introduction.
4- It is necessary to present the findings or results in the form of
tables related to content, mechanism, and outcome.
5- The study does not have a discussion section.
Author Response
1- The abstract needs to be edited because it lacks explanations about the study method and findings and conclusions.
- Thank you for your comments. The abstract has been rewritten. The revised manuscript has been reviewed by English language professional.
2- Telehealth is not synonymous with digital health, while in this study these two concepts were used interchangeably. Digital health is a type of electronic health based on artificial intelligence, mobile health, and molecular and cellular studies as well as precision medicine. It is suggested to use the term digital health because the focus of the study is on telehealth.
- Thank you for your comments. It is true that these terms use interchangeably in the literature. In our manuscript we emphases the term telehealth to give better and wider perspective to the casual readers of healthcare.
3- It is necessary to explain about the study method after the introduction.
- Thank you for your comments. Methodology has been explained in the revised manuscript.
4- It is necessary to present the findings or results in the form of tables related to content, mechanism, and outcome.
- Thank you for your suggestion. Now we reported findings in Table 1.
5- The study does not have a discussion section.
- Thank you for your valuable comments. As this is a review paper, we interpret the findings along with the results. We though that adding discussion section will enhance repetition. A separate discussion section can be included if the reviewer believed it would add value.
Reviewer 3 Report
Comments and Suggestions for Authors
This paper will provide a narrative review of telehealth solutions in COPD management.
The paper is well organized, and the length is appropriate. The title is chosen correctly, but the abstract doesn’t provide sufficient information to understand what to expect from the paper.
The results are poorly highlighted, and the discussion section is practically missing.
The limitations of the study should be specified.
The conclusion does not summarise all the issues discussed in the paper.
The references are relevant and correctly chosen, and related work is discussed and cited appropriately.
There are errors in references to figures and tables (See Row: 65, 80, 99, 106, 122, 147, 156)
What is “Traditional data analysis” in a digital context?
Fig 2 does not provide any clarification of the text.
However, I think the authors can improve the paper to reach the scientific standard required for publication.
Author Response
This paper will provide a narrative review of telehealth solutions in COPD management.
The paper is well organized, and the length is appropriate. The title is chosen correctly, but the abstract doesn’t provide sufficient information to understand what to expect from the paper.
- Thank you for your comments. Abstract has been rewritten, we hope it is clear and easy to follow in our revised manuscript.
The results are poorly highlighted, and the discussion section is practically missing.
- Thank you for your valuable comments. As this is a review paper, we interpret the findings along with the results. We though that adding discussion section will enhance repetition. A separate discussion section can be included if the reviewer believed it would add value.
The limitations of the study should be specified.
- Thank you for your comments. We have added limitations to the revised manuscript.
The conclusion does not summarize all the issues discussed in the paper.
- Thank you for your comments. Conclusion has been rewritten. We hope it is clear and precise now.
The references are relevant and correctly chosen, and related work is discussed and cited appropriately.
There are errors in references to figures and tables (See Row: 65, 80, 99, 106, 122, 147, 156)
- Thank you for your comments. Reference have been corrected.
What is “Traditional data analysis” in a digital context?
- We refer to storing and analysis data (A synchronization process). This including track and trace symptoms in individuals with COPD.
Fig 2 does not provide any clarification of the text.
- We felt that readers might benefit from having a diagram that explains how it works. We are happy to remove the figure if the reviewer feels that doing so would enhance the quality of the work.
However, I think the authors can improve the paper to reach the scientific standard required for publication.
- Thank you for your suggestions. We revised the manuscript to meet scientific standards. We hope it is clear and precise now.
Reviewer 4 Report
Comments and Suggestions for Authors
The article on Telehealth (TH) solutions for Chronic Obstructive Pulmonary Disease (COPD) offers an insightful and positive perspective on the potential benefits of utilizing telehealth in the management of this chronic condition.
I think author performed a comprehensive and well-written review which deserves publication, after addressing some minor concerns:
-
-Some abbreviations, such as "TH" for telehealth, are used without their full form upon first mention. It would be helpful to provide their full forms to ensure that readers who are not familiar with these acronyms can understand the context.
-
-Please proofread the article for punctuation and grammatical errors throughout the manuscript.
-
-In some sections, sentences are lengthy and complex, making it harder to follow the content. Consider breaking down long sentences for better readability.
- Please remove all the "Error! Reference source not found" from the manuscript and check for possible mistakes in references.
In conclusion, this article on telehealth solutions for COPD is highly valuable for its positive and holistic approach to improving the lives of individuals living with this chronic condition. By addressing various aspects, from context to mechanism and outcomes, and by recognizing the importance of patient engagement, it offers a promising perspective on the future of COPD care.
Comments on the Quality of English Languagemoderate editing required.
Author Response
The article on Telehealth (TH) solutions for Chronic Obstructive Pulmonary Disease (COPD) offers an insightful and positive perspective on the potential benefits of utilizing telehealth in the management of this chronic condition.
- Thank you for your comments. Your inputs have helped to improve the quality of the work. Very much appreciated.
I think author performed a comprehensive and well-written review which deserves publication, after addressing some minor concerns:
- Thank you for your comments. We addressed all minor corrections. Hope it is clear and precise now.
-Some abbreviations, such as "TH" for telehealth, are used without their full form upon first mention. It would be helpful to provide their full forms to ensure that readers who are not familiar with these acronyms can understand the context.
- Thank you for your comments. We have edited and correct all abbreviation mentioned in the revised manuscript.
-Please proofread the article for punctuation and grammatical errors throughout the manuscript.
- Thank you. We reviewed the revised manuscript to proofread the article for punctuation and grammatical errors.
-In some sections, sentences are lengthy and complex, making it harder to follow the content. Consider breaking down long sentences for better readability.
- We rephrased the context to be clear and easier for the casual readers. Hope it is clear now.
- Please remove all the "Error! Reference source not found" from the manuscript and check for possible mistakes in references.
- Thank you for the comments. All error references have been corrected in the revised manuscript.
In conclusion, this article on telehealth solutions for COPD is highly valuable for its positive and holistic approach to improving the lives of individuals living with this chronic condition. By addressing various aspects, from context to mechanism and outcomes, and by recognizing the importance of patient engagement, it offers a promising perspective on the future of COPD care.
- Thank you for the comments. We rephrased the conclusion to reflect the findings from our narrative review.
Reviewer 5 Report
Comments and Suggestions for Authors
Many thanks for inviting me to review this paper. This study summarized the TH intervention on COPD. I write my suggestions below and attached file.
I extend my sincere gratitude for extending an invitation to review the manuscript. The subject matter of this study is evidently aligned with the thematic purview of the esteemed Healthcare journal, renowned for its academic standing and discerning readership. Given the journal's stature, it is imperative that publications maintain a standard of excellence commensurate with the expectations of its audience.
While acknowledging the merit of the study, I find the issue of novelty to be somewhat equivocal. It is incumbent upon the authors to expound upon and persuade both readers and reviewers regarding the significance of their work. Noteworthy limitations in the study necessitate attention, leading me to question its suitability for publication in Healthcare. The augmentation of pertinent information is warranted for a more comprehensive understanding.
The fact that the study centered on how to apply the TH rather than the benefit of COPD leads to the need to revise the title in this direction.
The information about main problem related to COPD and TH, is rather short or missing. I think more information should be given in abstract about the rationale and need for TH.
The elucidation of the take-home message requires refinement, and I posit that a more detailed exposition of outcomes is essential. I recommend that the authors furnish additional details to underscore the essence of the take-home message.
I advise adherence to the MeSH headings for keywords to enhance the manuscript's indexing and discoverability.
The introduction section merits improvement, particularly in delineating distinctions from extant studies on the same subject. The rationale for engaging with this specific study amidst the plethora of investigations into the topic of interest remains unclear. A thorough explication of the study design and sampling methodology is imperative, as is a more explicit articulation of the take-home message.
This study needs to clearly articulate its unique contributions amid the abundance of existing research on the topic. Readers should be convinced of the study's distinctiveness, whether through innovative methodologies, fresh perspectives, or addressing gaps in current knowledge. The manuscript should succinctly convey why individuals in the field should invest their time in this particular study by highlighting its specific and valuable insights.
Did the authors preferred to use any of the guidelines or checklist like PRIMSA, STROBE checklist? Which would be more useful in terms of their study design.
Clinical outcomes of the given studies are not elaborated at all. As a reader I would like to see some solid evidence about the benefits of TH.
The conclusion should be elaborated in detail. In present manuscript the discussion is not evaluated enough the hypothesis. What are the future directions for further studies should be elaborated in the text?
The nexus between the study's objectives and its results necessitates meticulous elucidation. I advocate for strict adherence to the guidance provided for authors, particularly in relation to the journal's guidelines, notably those pertaining to the reference section.
Discrepancies in the inclusion of DOI numbers and page references require rectification to ensure uniformity and adherence to the journal's standards.
In conclusion, I express my earnest expectation that these constructive observations contribute positively to the refinement and enhancement of the manuscript, aligning it more closely with the standards befitting the Healthcare journal.

-
Author Response
Many thanks for inviting me to review this paper. This study summarized the TH intervention on COPD. I write my suggestions below and attached file.
I extend my sincere gratitude for extending an invitation to review the manuscript. The subject matter of this study is evidently aligned with the thematic purview of the esteemed Healthcare journal, renowned for its academic standing and discerning readership. Given the journal's stature, it is imperative that publications maintain a standard of excellence commensurate with the expectations of its audience.
- Thank you for your valuable comments. Your inputs have helped to improve the quality of the revised manuscript.
While acknowledging the merit of the study, I find the issue of novelty to be somewhat equivocal. It is incumbent upon the authors to expound upon and persuade both readers and reviewers regarding the significance of their work. Noteworthy limitations in the study necessitate attention, leading me to question its suitability for publication in healthcare. The augmentation of pertinent information is warranted for a more comprehensive understanding.
- Thank you for your comments. We revised the manuscript to be clearer and more precise for the readers.
The fact that the study centered on how to apply the TH rather than the benefit of COPD leads to the need to revise the title in this direction.
- Thank you for the comment. The title of this narrative review reflects the result of narrative review has been conducted.
The information about main problem related to COPD and TH, is rather short or missing. I think more information should be given in abstract about the rationale and need for TH.
- Thank you for the comment. We have revised the manuscript including the abstract to provide more information regarding the aim, methods, and results of this narrative review.
The elucidation of the take-home message requires refinement, and I posit that a more detailed exposition of outcomes is essential. I recommend that the authors furnish additional details to underscore the essence of the take-home message.
- Thank you for your valuable comments. The revised manuscript now includes a clear and concise take-home message for the readers.
I advise adherence to the MeSH headings for keywords to enhance the manuscript's indexing and discoverability.
- Thank you for the comment. We provide a search strategy as a supplementary file. MeSH terms have been applied.
The introduction section merits improvement, particularly in delineating distinctions from extant studies on the same subject. The rationale for engaging with this specific study amidst the plethora of investigations into the topic of interest remains unclear. A thorough explication of the study design and sampling methodology is imperative, as is a more explicit articulation of the take-home message.
- Thank you for your valuable comments. New references have been added to the introduction as well as corrections have been done to the reference list. The revised manuscript now includes a clear and concise take-home message for the readers.
This study needs to clearly articulate its unique contributions amid the abundance of existing research on the topic. Readers should be convinced of the study's distinctiveness, whether through innovative methodologies, fresh perspectives, or addressing gaps in current knowledge. The manuscript should succinctly convey why individuals in the field should invest their time in this particular study by highlighting its specific and valuable insights.
- Thank you for your valuable comments. The study contribution has been added to the revised manuscript. By reading the current study the reader will have a concise knowledge about various aspects, from context to mechanism and outcomes, as well as recognizing the importance of patient engagement in designing TH solutions which is offers a promising perspective on the future of COPD care.
Did the authors preferred to use any of the guidelines or checklist like PRIMSA, STROBE checklist? Which would be more useful in terms of their study design.
- Thank you for your comment. This is a narrative review not a systematic review so using PRISMA checklist will confuse the readers.
Clinical outcomes of the given studies are not elaborated at all. As a reader I would like to see some solid evidence about the benefits of TH.
- Thank you for the comments. We review the overall effect of using TH solutions on clinical outcomes. We did not specify the research question on clinical outcomes such as quality of life measures or physical activity measures. We agreed that the suggested question by the reviewer would be a create research question for future work.
The conclusion should be elaborated in detail. In present manuscript the discussion is not evaluated enough the hypothesis. What are the future directions for further studies should be elaborated in the text?
- We rephrased the conclusion in the revised manuscript.
The nexus between the study's objectives and its results necessitates meticulous elucidation. I advocate for strict adherence to the guidance provided for authors, particularly in relation to the journal's guidelines, notably those pertaining to the reference section.
- Thank you for your comments. We revised the manuscript to provide a deep understanding for the rational of doing this research as well as we improved the quality of the writing as suggested.
Discrepancies in the inclusion of DOI numbers and page references require rectification to ensure uniformity and adherence to the journal's standards.
- References list has been corrected.
In conclusion, I express my earnest expectation that these constructive observations contribute positively to the refinement and enhancement of the manuscript, aligning it more closely with the standards befitting the Healthcare journal.
- Thank you for you for your valuable comments. Your suggestions and comments have help in improve the readability and quality of the revised manuscript.
Round 2
Reviewer 2 Report
Comments and Suggestions for Authors
The content of the article has become much richer compared to the previous one, but there are still problems.
1- The justification that the review article prevents the discussion in a separate section is not logical. Even now, the results section lacks reasoning and inferences. Therefore, it is recommended to consider the discussion section in the article and make reasoning and inference about the results from the study of the texts.
2- Also, the conclusion section, which should reflect the total of the conclusions of the discussion section, should be rewritten. It is recommended to add the author's suggestions to the conclusion section.
Comments on the Quality of English LanguageIt's O.K
Author Response
Reviewer 2
The content of the article has become much richer compared to the previous one, but there are still problems.
1- The justification that the review article prevents the discussion in a separate section is not logical. Even now, the results section lacks reasoning and inferences. Therefore, it is recommended to consider the discussion section in the article and make reasoning and inference about the results from the study of the texts.
- Thank you for your valuable comments. We revised the results sections in the manuscript. We also add discussion section to the manuscript. Hope it is clear now.
2- Also, the conclusion section, which should reflect the total of the conclusions of the discussion section, should be rewritten. It is recommended to add the author's suggestions to the conclusion section.
- Thank you for your comment. We added both discussion and conclusion.
Reviewer 3 Report
Comments and Suggestions for Authors
The revised version of the paper has been improved.
The paper meets the requirements for a scientific article published in a quality journal.
Author Response
Thank you for your comments. Very much appreciated.
Reviewer 5 Report
Comments and Suggestions for Authors
Dear Editor
Please find below our views on the article you sent to me for my evaluation.
I believe that the authors' responses and revisions to the previous peer reviews have contributed significantly to the quality of the manuscript. I would like to state that I am convinced by the revisions particularly in the hypothesis, method and discussion sections. I would like to state that the acceptance of the article in this form is appropriate and I congratulate the authors.
Yours sincerely
Author Response
- Thank you for your comments. Very much appreciated.